# Evolution of the Second-Phase Particles and Their Effect on Tensile Fracture Behavior of 2219 Al-*x*Cu Alloys

**Daofen Xu [1,2,3,4], Kanghua Chen [1,4,*], Yunqiang Chen [1,4] and Songyi Chen [1,4]**

1    Light Alloy Research Institute, Central South University, Changsha 410083, China;
     xu_daofen@126.com (D.X.); shuracyq@163.com (Y.C.); sychen08@csu.edu.cn (S.C.)
2    Department of Mechanical Engineering, Guilin University of Aerospace Technology, Guilin 541004, China
3    Guangxi Colleges and Universities Key Laboratory of Robot & Welding,
     Guilin University of Aerospace Technology, Guilin 541004, China
4    National Key Laboratory of Science and Technology for National Defence on High-Strength Structural
     Materials, Central South University, Changsha 410083, China
*    Correspondence: khchen@csu.edu.cn; Tel.: +86-731-8883-0714

**Abstract:** In this study, the continuous evolution of the second-phase particles across as-cast, homogenization, multi-directional forging (MDF), and solution-aging treatment and their effect on tensile fracture behavior of 2219 aluminum alloys with different Cu contents was examined by optical microscopy (OM), scanning electron microscopy (SEM), and tensile tests. The results showed that the microstructure of as-cast 2219 aluminum alloy consisted of the $\alpha$-Al matrix, $Al_2Cu$ coarse phase, and Fe-rich impurity phase. Severe segregation of Cu existed, and eutectic networks can be observed in the ingot. With an increase in Cu content, the eutectic networks became coarsen and thicker. During the complex improved process, the refinement mechanisms were fragmentation, dissolution, and diffusion of $Al_2Cu$ particles. Most fine $Al_2Cu$ particles were fully dissolved into the matrix and partial coarse particles were still retained after solution-aging treatment. Thus, the elongations of all the samples, undergoing solution treatment followed by water quenching, increased evidently. Then, the elongations decreased slightly due to the increase of precipitates. The fractography analysis of peak aged condition samples indicated that the fracture mode was diverted from a typical inter-granular fracture to a mainly trans-granular fracture with increase in Cu content from 5.56% to 6.52%. Fracture initiation mainly occurred by original microcrack propagation and microvoid nucleation at the coarse constituents.

**Keywords:** 2219 aluminum alloys; constituents; tensile fracture behavior; cracks nucleation and propagation

## 1. Introduction

Because of its high specific strength, excellent corrosion resistance, good machinability, and sound weldability, the 2219 aluminum alloy has been extensively used to fabricate propellant tank of large launch vehicles, such as Saturn 5 and Long March 5 series launch vehicle [1,2]. The Cu content in this alloy (5.8–6.8%) exceeds its solubility limit in Al (5.65%). Thus, $Al_2Cu$ constituent can be formed readily [3]. Although the low melting point of $Al_2Cu$ eutectic phase can reduce the tendency of hot cracking during the welding, it brings some adverse effects. Xu et al. [4] studied the segregation of Cu across the ultra-large 2219 Al alloy ingot and found that the coarse $Al_2Cu$ particles could not be completely dissolved even after a complex thermo-mechanical treatment (TMT) process, and the un-dissolved particles were easily subjected to stress concentration during deformation and acted as

crack initiation sites, thus decreasing tensile ductility. Huang et al. [5] investigated the effect of the shape and size of particles on void nucleation mechanism and found that coarse particles with a high shape factor were concluded to be preferable for inducing the formation of cracks and accelerating the coalescence of micro-cracks. Yu et al. [6] reported that coarse $Al_2Cu$ particles can be elongated during the ring rolling process, which could be the main reason for anisotropy of mechanical properties. In addition, $Al_2Cu$ particles presented a cathodic nature with respect to $\alpha$-Al matrix [7–9]. Initially, corrosion of the $\alpha$-Al matrix occurred at $\alpha$-Al matrix/particles interface and later the spalling of $Al_2Cu$ particles were found with the prolonging of the corrosion time. Correspondingly, 2219 aluminum alloys exhibited a superior corrosion resistance when $Al_2Cu$ phase was in $\alpha$-Al solid solution, according to Zhou et al. [10].

Except for $Al_2Cu$ coarse particles, $Al_7Cu_2Fe$ or $Al_7Cu_2$(Fe, Mn) impurity intermetallics may also be observed in 2219 aluminum alloy. The form of the Fe-rich intermetallic phase is related to composition of the alloys, cooling rate, and processing method, which has a similar influence on $Al_2Cu$ eutectic phase [11–14]. According to the results of our ongoing research, the impurity Fe-rich particles are not dissolved into $\alpha$-Al matrix and can also act as crack initiation sites during deformation, which is to further deteriorate the mechanical properties. Thus, Fe purification is an effective approach to preserve the fracture resistance. However, it is paradoxical to the utilization of recycled aluminum alloys. In addition, impurity Fe may also originate from melting equipments.

Note that the coarse particles were modified after introducing ultrasonic melt treatment [15,16]. It was found that the dissolution of Cu into $\alpha$-Al matrix increased to 4.3% and the segregation degree of Cu was weakened at the index of 0.2~0.15, and the mean size of $Al_7Cu_2Fe$ intermetallics decreased from 70 to 24 $\mu$m. Further, the refined $\alpha$-Al grains, together with the even distribution of solute, decreased the formation of coherent coarsen eutectic skeleton $Al_2Cu$ phase and increased the precipitation of fine $Al_2Cu$ particles inside the grains [17]. Therefore, the area fraction of the coarsening eutectic phase was reduced to a fairly low value.

In fact, $Al_2Cu$ and Fe-rich eutectic phases cannot be completely eliminated using heat treatment like homogenization and solution treatment. So, severe plastic deformation (SPD) processes [18–21], such as equal channel angular pressing (ECAP), accumulative roll bonding (ARB), high-pressure torsion (HPT), and multi-directional forging (MDF), have been extensively investigated. Among these SPD processes, MDF exhibits the enormous potential for aircraft field because it is a simple and cost-effective method to produce large bulk samples. Except for the grain refinement, coarse primary particles can also be refined and partly eliminated during MDF [21–23]. The main reason was that the dynamic fragmentation of primary particles could occur in forging at a low temperature, while the dissolution and diffusion of solute Cu atom increased with an increase in forging temperature. It is advisable that the refinement is associated with the characteristics of coarse particles, such as their shape, size, and distribution. Therefore, the coarse particles are apt to cleave during MDF. In addition, decohesion at the particles-matrix interface and dehiscence on primary brittle constituents was formed readily under service loading. However, limited studies have been concerned about the effect of characteristics of the second-phase particles on tensile fracture behavior. Therefore, this paper aims to investigate the evolution of the second-phase particles in 2219 Al-$x$Cu alloys during a complex TMT process and their effect on mechanical properties, with the aim to study the tensile fracture behavior induced by the second-phase particles.

## 2. Materials and Methods

The 2219 aluminum alloys with different Cu contents were melted in an electrical furnace and degassed by $C_2Cl_6$. The molten metal was maintained at 720 to 730 °C for 30 min before being poured into a cylindrical steel mold with a dimension of $\varphi$100 mm. The mold was preheated to approximately 250 °C before pouring. During solidification, the cooling water was sprayed on the ingot surface until the casting was completely solidified. The chemical compositions were measured by inductively coupled plasma optical emission spectroscopy and the results were listed in Table 1.

**Table 1.** Chemical compositions of experimental alloys, wt.%.

| Sample No. | Cu | Mn | Fe | Si | Mg | Zn | Ti | V | Zr | Al |
|---|---|---|---|---|---|---|---|---|---|---|
| 2219-5.56Cu | 5.56 | 0.362 | 0.10 | 0.0005 | ≤0.02 | ≤0.1 | ≤0.1 | 0.05–0.15 | 0.10–0.25 | Bal. |
| 2219-6.15Cu | 6.15 | 0.356 | 0.09 | 0.0005 | ≤0.02 | ≤0.1 | ≤0.1 | 0.05–0.15 | 0.10–0.25 | Bal. |
| 2219-6.52Cu | 6.52 | 0.358 | 0.09 | 0.0005 | ≤0.02 | ≤0.1 | ≤0.1 | 0.05–0.15 | 0.10–0.25 | Bal. |

The ingots were homogenized in a resistance air furnace at 525 °C for 24 h. All the samples were subjected to multiple MDF at 450 °C with a compression speed of 6mm/s by using a numerically controlled hydraulic press (YH27-500T; Hefei Forging Machine Co. Ltd., Hefei, China). BN was used as a lubricant during MDF. The samples were repeatedly forged by changing the loading direction through 90° (*X*, *Y*, *Z*), which included 3 passes of upset forging along different axes (the compression deformation was kept at 50%), 3 passes of stretch forging along with the un-deformed axes, and 1 pass of shape forging, then quenched in water at ambient temperature immediately. After MDF, the samples were solution treated at 545 °C for 4 h and quenched in water of room temperature. Thirdly, the process of MDF was repeated and was solution treated at 537 °C for 4 h. Finally, the samples were aged at 165 °C for 24 h and cooled in water. The interval time between solution treatment and artificial aging did not exceed 2 h.

The geometry of tensile specimen was designed according to the standard of GB/T228-2002 and all the tests were carried out using an Instron electronic universal testing machine (Model 3369, Instron Co., Canton, MA, USA) at room temperature with a constant speed of 2 mm/min. For each condition, three samples were made to obtain an average value. The samples were prepared by successive grinding with silicon carbide papers #400, #800, #1500, #2000 and then mechanically polished. Subsequently, samples were etched with Keller's reagent. The average grain size was measured by a linear intercept method (ASTM E112-10). The microstructures and fracture morphologies were observed using an optical microscopy (OM; DM4000M, Leica Microsystems, Wizz, German) and a scanning electron microscopy (SEM; Nova Nano SEM230, FEI Co., Hillsboro, OR, USA) equipped with an energy dispersive spectrometer (EDS).

## 3. Results

### 3.1. Microstructures of As-Cast and Homogenized Alloys

Figure 1 shows the OM micrographs of as-cast 2219 aluminum alloys with different Cu contents. They exhibited a typical dendrite structure in the ingots and the dendrite segregation increased with increase in Cu contents. The average grain size increased gradually from 443.6 ± 10 μm to 509.4 ± 10 μm with an increase in Cu content from 5.56% to 6.52%. The microstructure of 2219 aluminum alloy consisted of α-Al matrix and Cu-rich intermetallic phases. Spherical particles were mainly distributed inside the grains, and the number density of these particles decreased with increase in Cu contents.

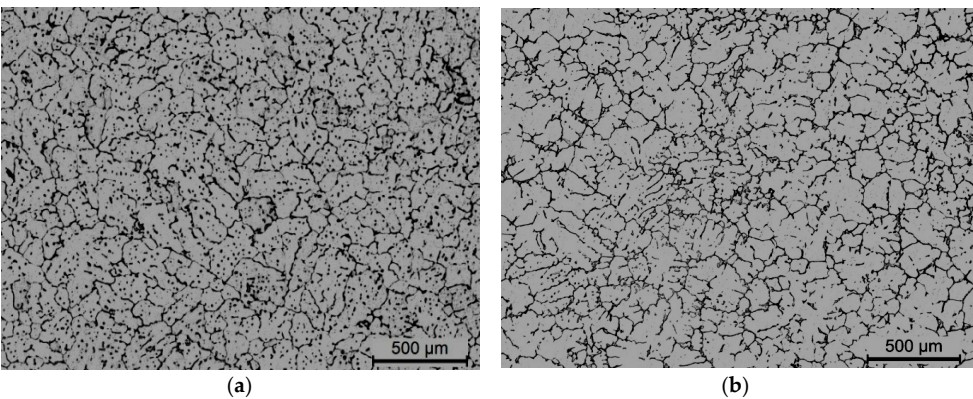

(a)  (b)

**Figure 1.** *Cont.*

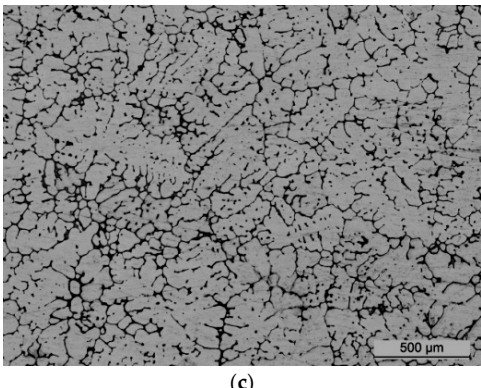

(**c**)

**Figure 1.** Optical micrographs of as-cast 2219 aluminum alloys with different Cu contents: (**a**) 5.56%, (**b**) 6.15%, (**c**) 6.52%.

Figure 2 shows the OM micrographs of as-cast 2219-6.15Cu and 2219-6.52Cu alloys. It can be seen that the eutectic phases segregated seriously at grain boundaries. As illustrated in Figure 2b, two different phases precipitated along the grain boundary: a white $Al_2Cu$ phase (Point A) having tri-pole junction morphology and a grey $Al_7Cu_2(Fe, Mn)$ phase (Point B) attached to the edge of $Al_2Cu$ phase, which was further confirmed by EDS (seen in Figure 3). The white $Al_2Cu$ phase (Point C) also precipitated inside the grains. The lamellar eutectic phase in Figure 2c was dramatically thicker than that in Figure 2a. The distribution of main elements was further confirmed by line scanning, as seen in Figure 2d. It is evident that the segregation of Cu is much greater than that of Fe and Mn along the grain boundary, leading to the coarsening eutectic networks.

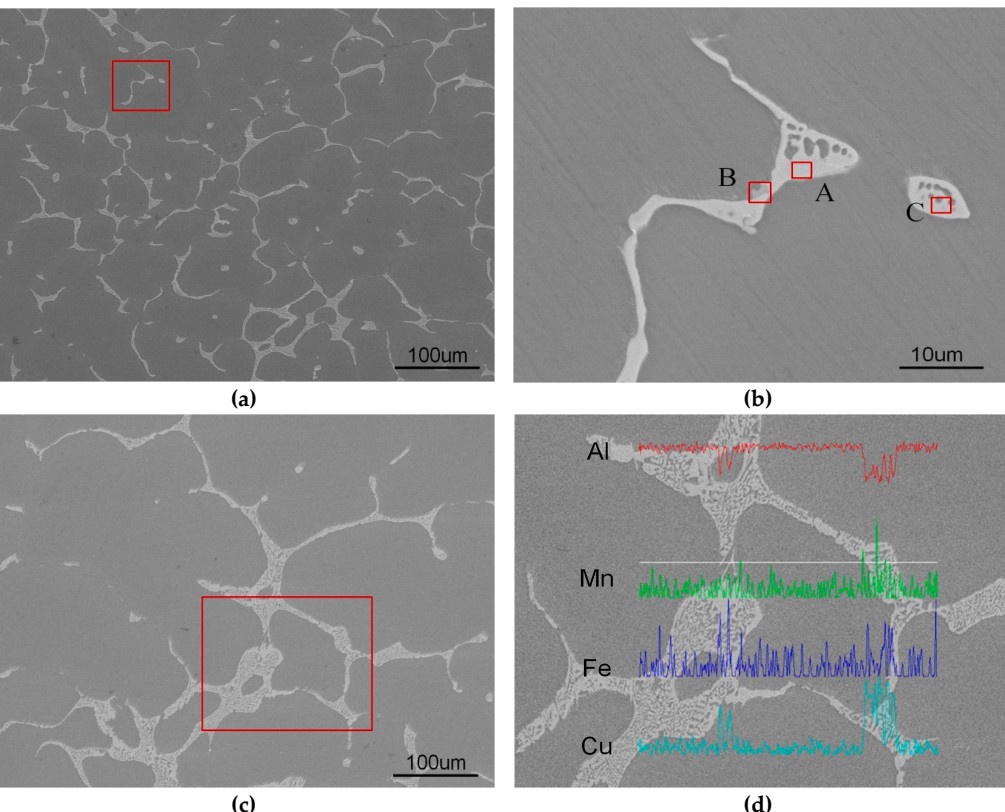

**Figure 2.** Scanning electron microscopy micrographs of as-cast 2219-6.15Cu and 2219-6.52Cu alloys: (**a,b**) 2219-6.15Cu alloy; (**c,d**) 2219-6.52Cu alloy.

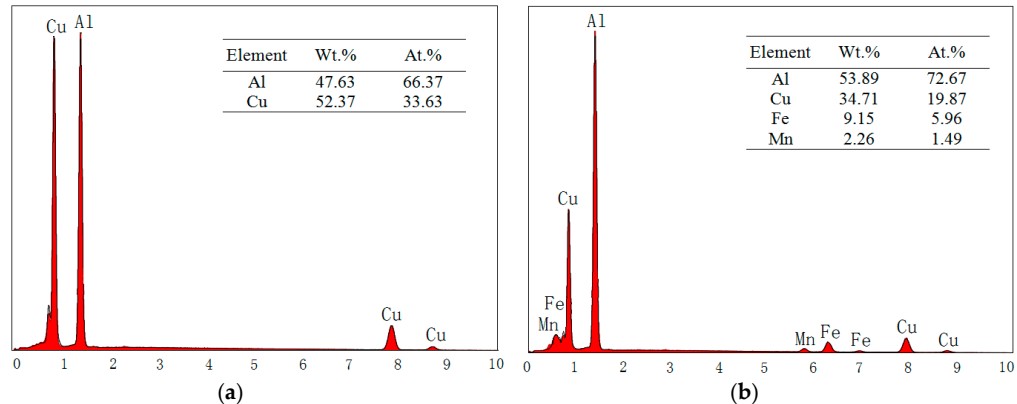

**Figure 3.** The second-phase particles component detected by energy dispersive spectrometer: (**a**) Point A and C ($Al_2Cu$); (**b**) Point B ($Al_7Cu_2$(Fe, Mn)).

Figure 4 shows the micrographs of homogenized 2219 aluminum alloys with different Cu contents. It could be seen that the dendrite features were not completely eliminated in homogenized alloys, whereas, the grain boundaries became thinner and clearer. As illustrated in Figure 4a, most constituents, which were revealed inside the grains in as-cast condition, were dissolved into the matrix. For 6.15% content of Cu, except for a limited number of constituents that remained inside the grains, the residual constituents became smaller and distributed discontinuously along the grain boundaries, as can be seen in Figure 4b,d. However, the dendrite network structure existed obviously, and most residual constituents were distributed continuously along the grain boundaries when Cu content was increased to 6.52%, as shown in Figure 4c.

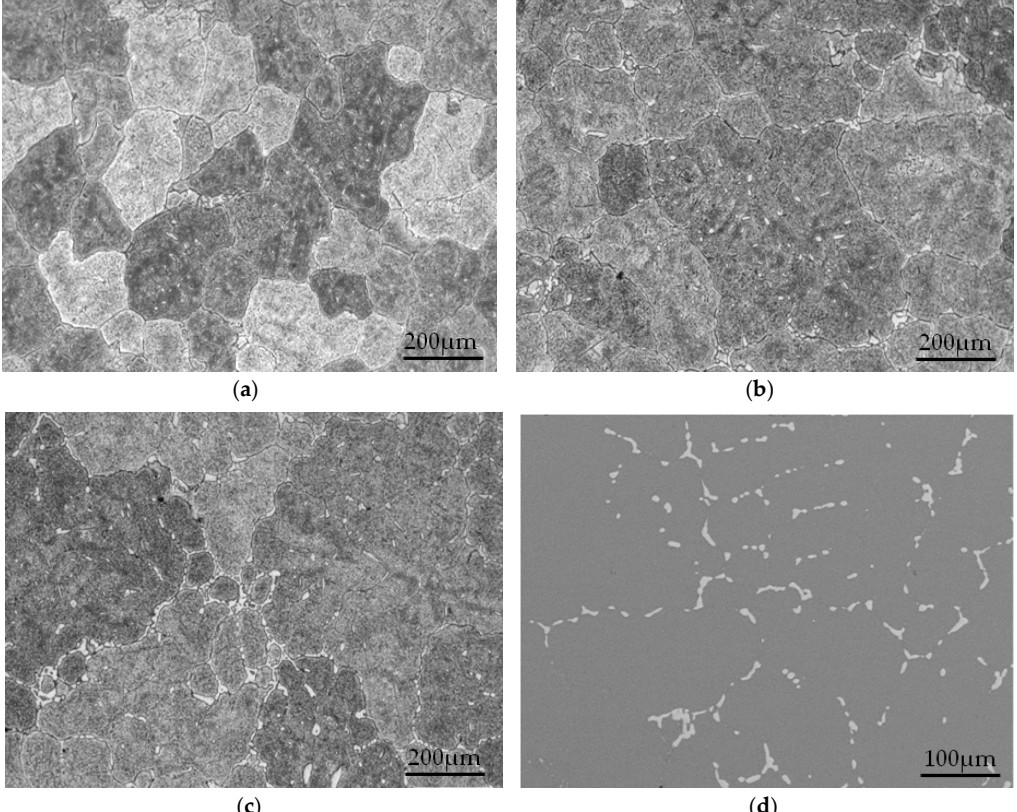

**Figure 4.** Micrographs of homogenized 2219 aluminum alloys with different Cu contents: Optical images (**a**) 5.56%, (**b**) 6.15%, (**c**) 6.52%, and (**d**) SEM image of 6.15%.

## 3.2. Microstructures after MDF

Figure 5 shows the micrographs of 2219 aluminum alloys with different Cu contents processed by MDF at 450 °C. It can be seen that the constituents clearly decreased and the residual particles were unevenly distributed. For 5.56% content of Cu, the coarse particles were spheroidized. With an increase in Cu content, spheroidized particles gradually reduced, while the coarse particles exhibited an elongated shape. Comparing to as-homogenized alloys, the area fraction of un-dissolved coarse particles decreased significantly. This phenomenon can be attributed to the dissolution of Cu in the matrix during MDF at high temperatures. The morphology of un-dissolved coarse particles in 2219-6.52Cu alloy was observed by SEM at a high magnification, as shown in Figure 5d. As marked with arrows, some coarse particles were fragmented incompletely, and visible cracks formed on primary coarse particles. The reason is that coarse particles with a high shape factor are subjected to high levels of stress concentration, leading to dehiscence on these particles. The constituents cleaved along different directions, because of the large strains with multiple directions. Generally speaking, the larger the particles, the easier to cleave.

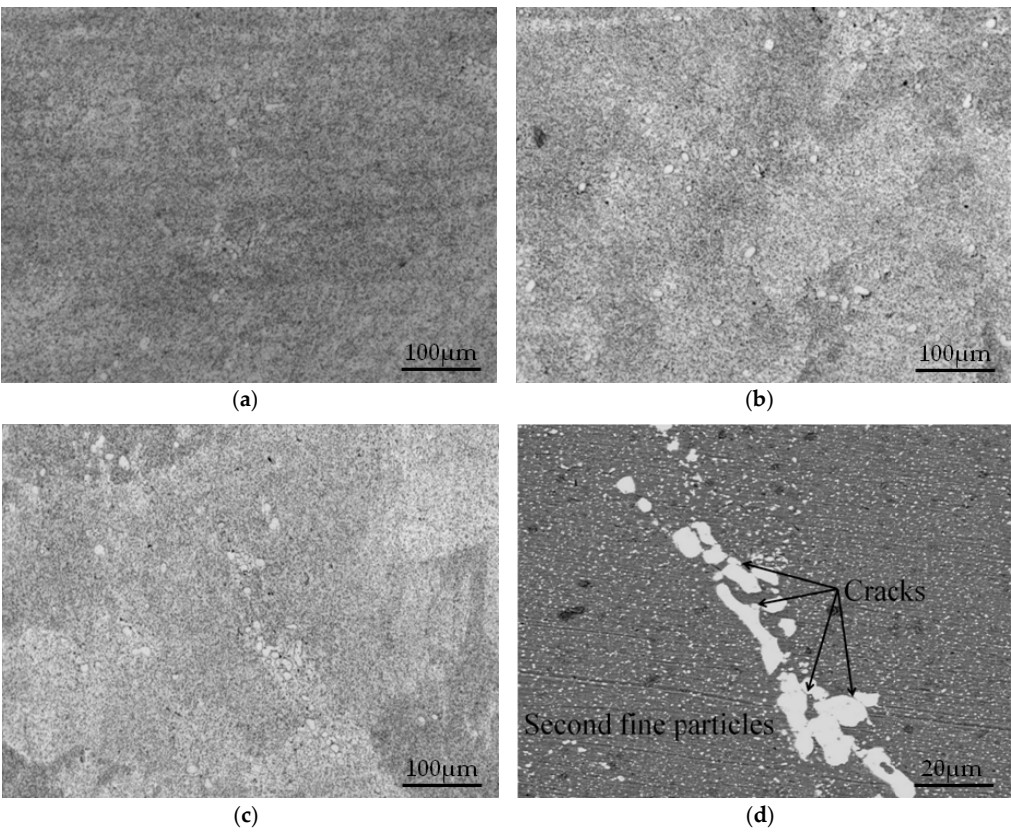

(a)　(b)

(c)　(d)

**Figure 5.** Micrographs of 2219 aluminum alloys with different Cu contents processed by multi-directional forging (MDF): Optical images (**a**) 5.56%, (**b**) 6.15%, (**c**) 6.52%, and (**d**) SEM image of un-dissolved coarse particles in 2219-6.52Cu alloy.

Many fine $Al_2Cu$ particles, with 0.5 um average size, were evenly distributed in the matrix, as shown in Figure 5d. These dispersed particles formed during water quenching after MDF. The $Al_{20}Cu_2Mn_3$ and $Al_3Zr$ particles are two other types of dispersoids in 2219 alloys, which formed during homogenization [24,25]. Fine and uniformly distributed dispersed particles can effectively pin dislocation, providing more nucleation positions for the formation of aged precipitated phase θ′.

## 3.3. Microstructures after Solution and Aging Treatment

Figure 6 shows the optical micrographs of aging-2219 aluminum alloys with different Cu contents. It can be seen that all of the alloys had coarse and inhomogeneous equiaxed grains. As illustrated

in Figure 6a,c,e, the mean grain size for 2219-5.56Cu was 280 ± 10 μm, which increased to 353.8 ± 10 μm and 469.2 ± 10 μm for 2219-6.15Cu and 2219-6.52Cu, respectively. The grain size was related to the second-phase particles. Dynamic fragmentation, dissolution, and diffusion of Al$_2$Cu particles were expected to occur during MDF at this experimental condition. Thus, the deformation energy was low for recovery, resulting in a limited number of nucleation sites for static recrystallization during the solution treatment. However, the high-density dislocations formed around coarse particles and dynamic recrystallization could have occurred during MDF. Thus, the residual coarse particles could be observed inside the grains, as shown in Figure 6b,d,f. Compared to the samples processed by MDF, the amount and size of coarse particles clearly decreased after solution treatment. The insoluble Fe-rich phases were also observed, as marked by arrows.

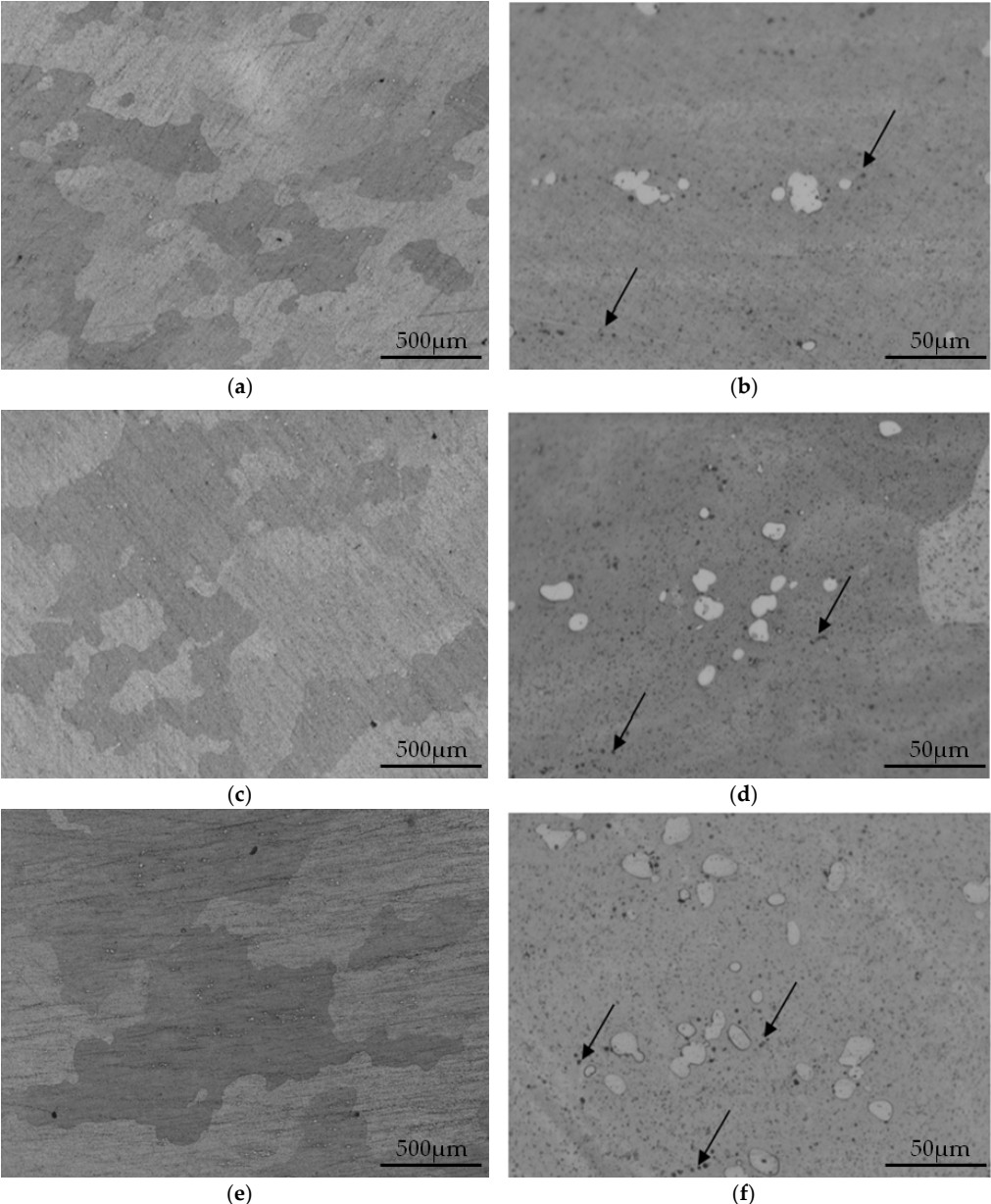

**Figure 6.** Optical micrographs of aging-2219 aluminum alloys with different Cu contents: (**a**,**b**) 5.56%, (**c**,**d**) 6.15%, (**e**,**f**) 6.52%.

Figure 7 shows the grain boundary structure of 2219 Al-*x*Cu alloys after solution treatment followed by water quenching. It can be seen that quench-induced equilibrium particles preferentially

nucleated along the grain boundaries. Meanwhile, the number and size of grain boundary precipitates increased with an increase in Cu content. For 2219-5.56Cu alloy, a small number of fine particles, less than 0.5 μm in size, existed along the grain boundary. For 2219-6.52Cu alloy, the fine particles, in the size range of 2–8 μm, also mainly occupied the grain boundary as well. The fine particles, that is, $Al_2Cu$, were confirmed by map scanning, as shown in Figure 8.

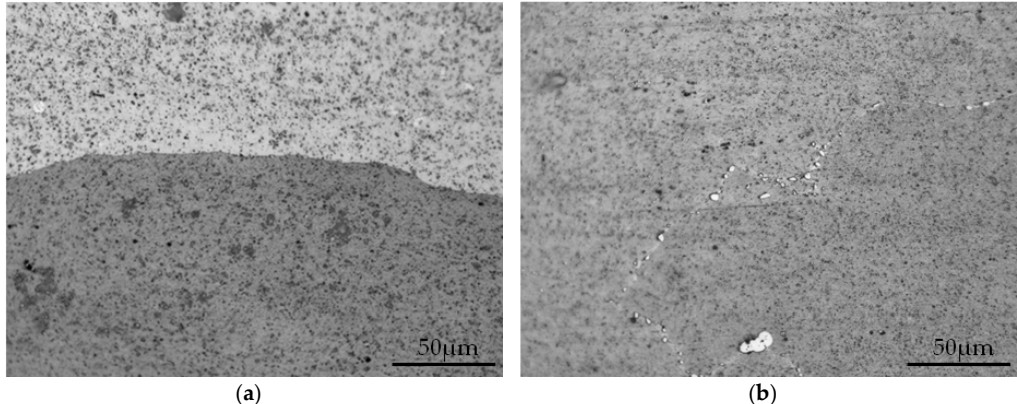

(a)  (b)

**Figure 7.** Optical micrographs of solution treatment followed by water quenching 2219 Al-*x*Cu alloys: (**a**) 5.56%, (**b**) 6.52%.

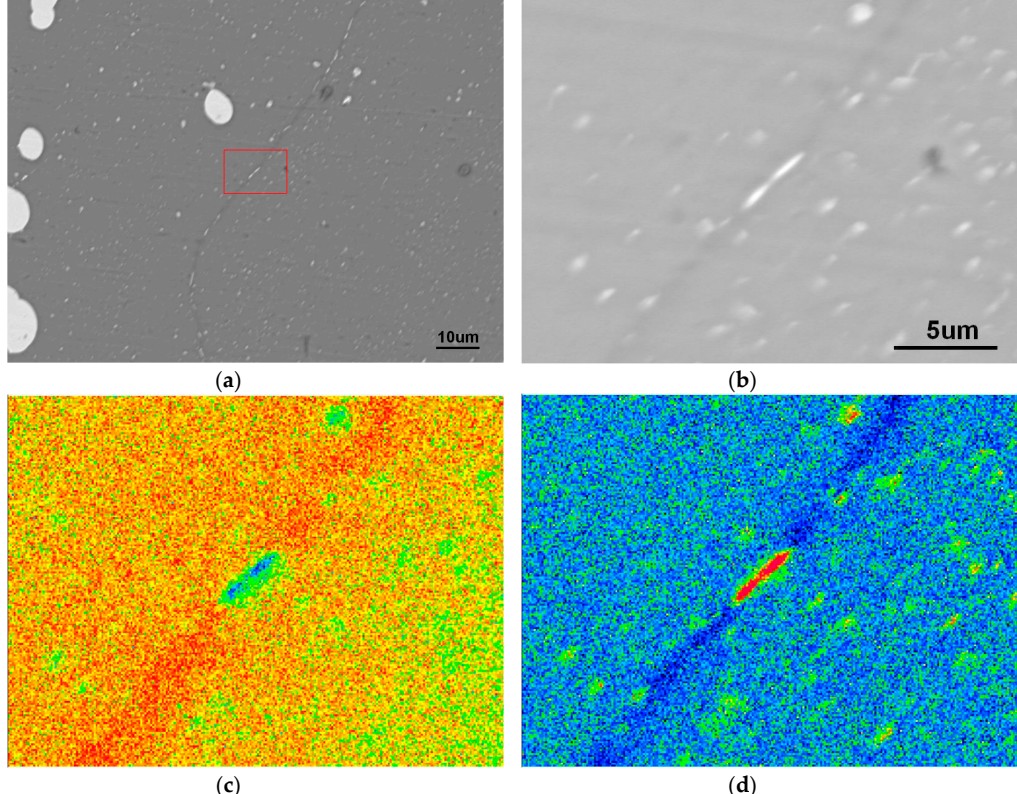

(a)  (b)

(c)  (d)

**Figure 8.** SEM micrographs and element distribution of solution treatment followed by water quenching 2219Al-6.52Cu alloy: (**a**,**b**) SEM image, (**c**) Al distribution, (**d**) Cu distribution.

### 3.4. Mechanical Properties and Tensile Fracture Morphology

Figure 9 shows the mechanical properties of 2219 aluminum alloys with different Cu contents. Under different processes, the ultimate tensile strength (UTS) and yield strength (YS) were always on the upward trend with the increase of Cu content. However, the elongation (EL) increased with

the process from homogenization to solution followed by water quenching and decreased with peak aged condition. The as-homogenized samples presented relatively low UTS/YS/EL values, i.e., 225.96 MPa/102.34 MPa/12.91% for 2219-5.56Cu alloy, 204.78 MPa/89.72 MPa/9.23% for 2219-6.15Cu alloy, and 182.56 MPa/76.58 MPa/4.79% for 2219-6.52Cu alloy. Adding complexity, the TMT process significantly increased the UTS/YS/EL values by at least 70 MPa, 100 MPa, and 6.3%, respectively. In terms of the peak aged samples, the UTS/YS values increased to 423.33/316.95 MPa for 2219-5.56Cu alloy, 429.67/330.38 MPa for 2219-6.15Cu alloy, 373.05/299.41 MPa for 2219-6.52Cu alloy, whereas the EL values were reduced to 18.27%, 13.58%, and 7.75%, respectively. The huge difference in mechanical properties with different Cu contents and processes was because of the variation of different $Al_2Cu$ second-phase particles.

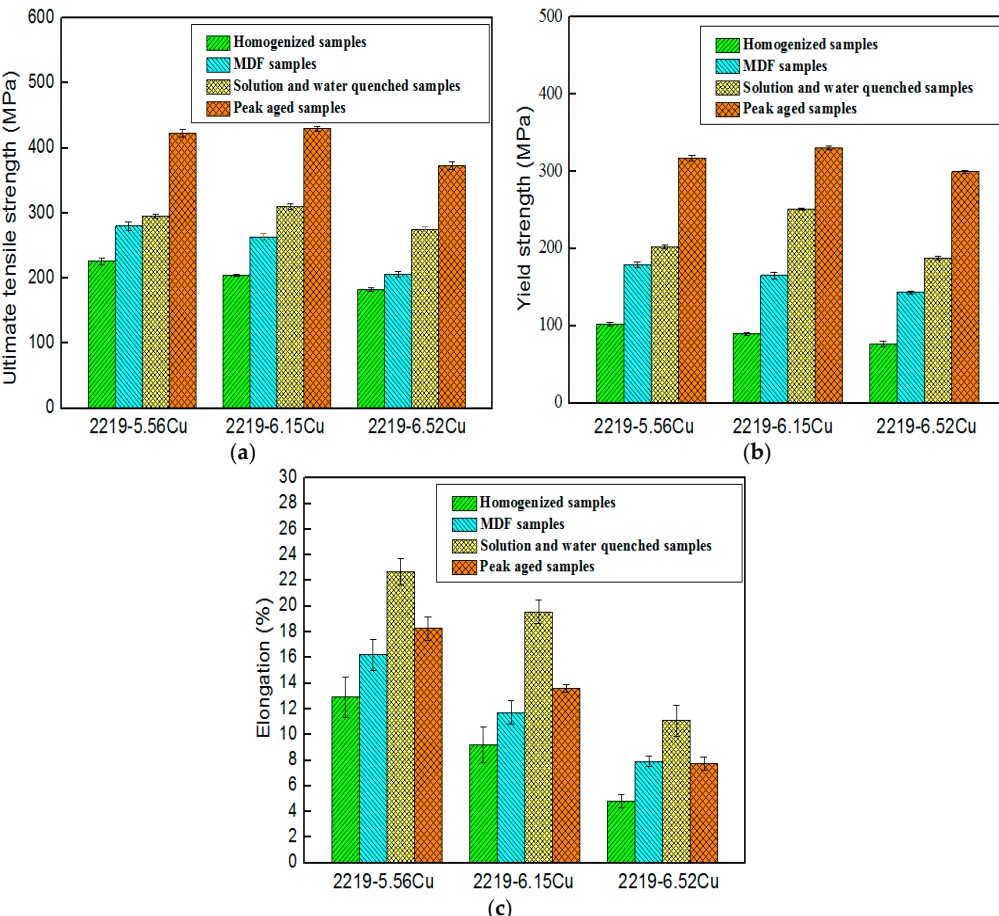

**Figure 9.** Mechanical properties of 2219 aluminum alloys with different Cu contents: (**a**) The ultimate tensile strength, (**b**) the yield strength, (**c**) elongation.

Figure 10 shows the tensile fracture morphology of the peak aged 2219 aluminum alloys with different Cu contents. It can be seen that the fracture mode was diverted from a typical inter-granular fracture to a mainly trans-granular fracture with increase in Cu content from 5.56% to 6.52%. A large amount of big and deep voids formed by inter-granular fracture were observed in 5.56Cu specimen. This characteristic was caused by the grain boundary in high levels of stress concentration. Some dimples that formed around the fine particles were considerably shallow and equally distributed in the rough region (marked by ellipse in Figure 10a), as shown in Figure 10b. As shown in Figure 10c,d, some tiny tearing ridges and a bimodal dimple size distribution can be observed in the 6.52Cu specimen. The fine dimples were formed around the fine particles and occupied the ligaments between the primary dimples, while the primary dimples, caused by the fracture of coarse particles, were obviously larger and deeper. With a careful observation in Figure 10d, dehiscence on primary coarse particles or

decohesion at the particles-matrix interface existed in 6.52Cu alloy. This observation was in line with the fact that the primary coarse particles suffer a higher stress concentration than the fine dispersoid particles during plastic straining [26]. In brief, the coarse particles can act as crack initiations, thus reducing the ductility of 2219 Al alloy. Conversely, the finer particles can decrease the form and accumulation of microvoids, improving the ductility of the alloy.

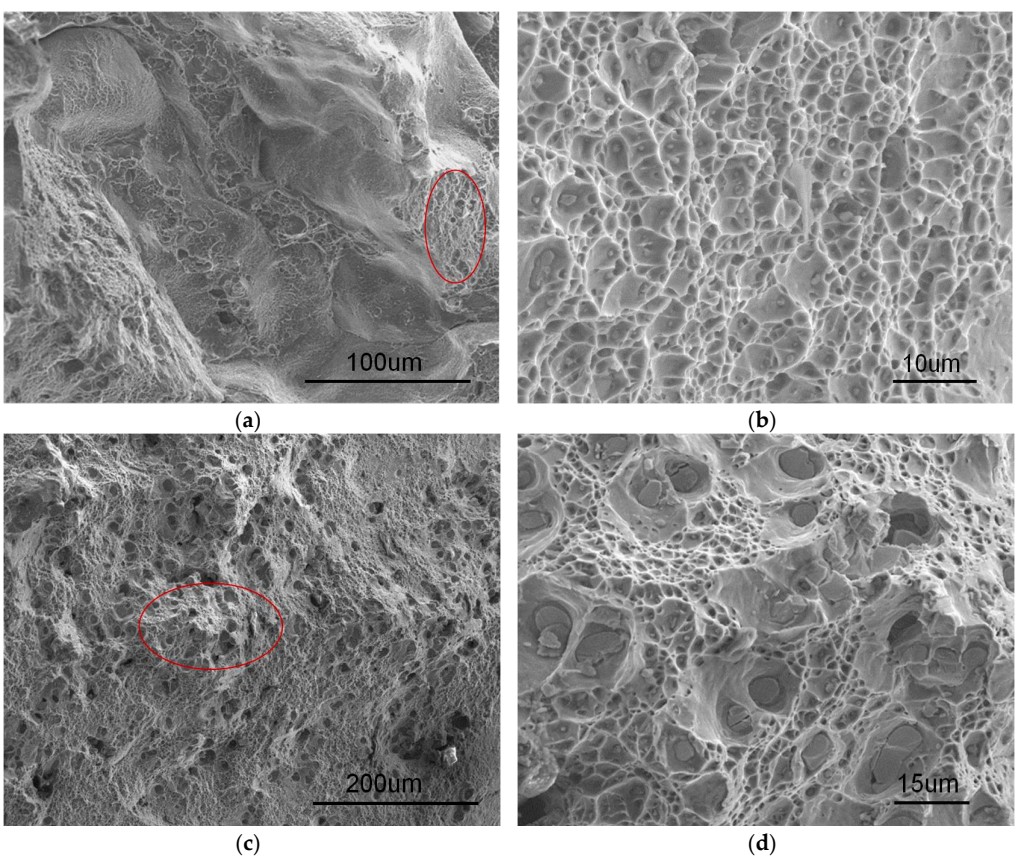

**Figure 10.** Tensile fracture morphology of 2219 aluminum alloys with different Cu contents: (**a,b**) 5.56Cu; (**c,d**) 6.52Cu.

## 4. Discussion

### 4.1. Refinement Mechanisms for the Second-Phase Particles

According to the Al-Cu-Mn-(Fe) alloy phase diagram (seen in Figure 11), the main phases including $Al_2Cu$, $Al_{20}Mn_3Cu_2$ and $Al_7Cu_2Fe$ exist in equilibrium composition [27]. Whereas, under real solidification conditions, the phase composition of as-cast alloys may deviate from the equilibrium composition, as shown in Figure 2. The supersaturated solid solution of Mn in the $\alpha$-Al matrix was formed, and some Mn bound to $Al_7Cu_2Fe$ phase to form $Al_7Cu_2(Fe, Mn)$ phase during solidification. So, the main phases in as-cast 2219 aluminum alloys would be $Al_2Cu$ and $Al_7Cu_2Fe/Al_7Cu_2(Fe, Mn)$. The $Al_2Cu$ eutectic veined at dendritic cell boundaries, while Fe participated in both eutectic and peritectic reaction. Therefore, the rod-like $Al_7Cu_2Fe/Al_7Cu_2(Fe, Mn)$ phases were distributed across the dendrite or attached to the edge of $Al_2Cu$ phase [28]. The $Al_2Cu$ phase was not completely dissolved, while the $Al_7Cu_2Fe/Al_7Cu_2(Fe, Mn)$ phase was not dissolved during the conventional solution treatment, because the contents of Cu and Fe in 2219 aluminum alloy were higher than their maximum solubility in Al. In fact, the larger the size of primary particles, the lower their solubility. In addition, high levels of stress concentration can be induced easily at coarse constituents, leading to dehiscence of constituents or decohesion at the particles-matrix interface, thereby deteriorating the mechanical property of alloys. So, it is necessary to refine the coarse constituents.

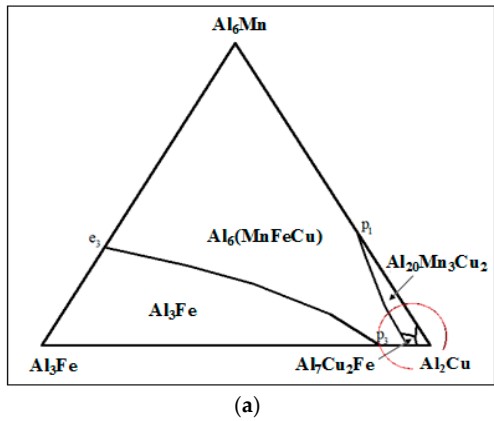 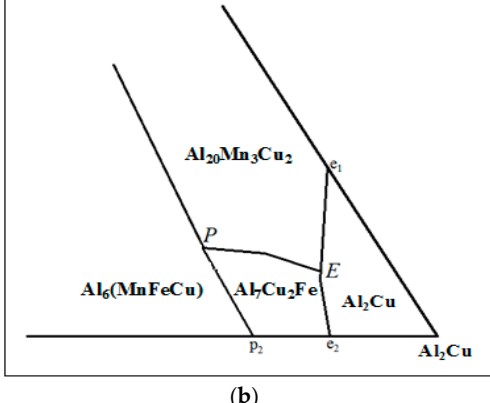

(a)  (b)

**Figure 11.** Phase diagram of intermetallics Al-Cu-Fe-Mn system alloy: (**a**) Polythermal projection of the solidification face in the Al corner; (**b**) enlarged view in the $Al_2Cu$ side.

Based on all the experimental results above, the continuous evolution of the second-phase particles across as-cast, homogenization, MDF, solution treatment, and water quenching followed by age treatment are summarized in Figure 12. As shown in Figure 12a, except for some spherical particles and needle-shaped precipitates (formed on slow cooling), the eutectic network mainly existed. After homogenization, just partial eutectic network and spherical particles were undissolved (seen in Figure 12b).

According to the application requirements of billet, complex TMT processes including MDF and solution treatment in turn was employed to refine the primary coarse particles (seen in Figure 12c–e). The main refinement mechanisms of the second phases are fragmentation, dissolution, and diffusion. Therefore, the refinement of the residual coarse phases is related to the process parameters, such as MDF temperatures, single-pass deformation, forging passes, and solution treatment. During the MDF, dislocation pileups occur at $Al_2Cu$ particles and matrix interface, leading to the formation of the compressive stress on the primary coarse particles (seen in Figure 12c, as marked by the circle in Figure 12b). If the stress is greater than its strength limit, that is, $|\sigma_1-\sigma_3| \geq [\sigma]$, the particle can be fragmented [29], where $|\sigma_1-\sigma_3|$ is the maximum principal stress deviator and $[\sigma]$ is the yield strength of the material. $|\sigma_1-\sigma_3|$ is related to the shape of the particle. The needle-like or rod-like second particle with a high value of $|\sigma_1-\sigma_3|$ is easier to fragment. Thus, the uniform distribution of fragments is from the sustaining increase of deformation (seen in Figure 12d). However, the coarse $Al_2Cu$ particles of the large size cannot be completely fragmented into small particles. Thus, visible cracks through the whole $Al_2Cu$ particle were formed, as shown in Figure 12e. Hence, these cracks can speed up the failure process in the subsequent tensile test, as shown in Figure 9c.

He et al. [21] applied the MDF to 2219 Al-Cu alloy and found that the dynamic fragmentation of $Al_2Cu$ particles was expected to exist at a low temperature, while the second-phase particles could dissolve and diffuse into the aluminum matrix at high temperatures. Meanwhile, the dissolution ability and diffusion rate were increased with increase in deformation temperature and deformation degree. In addition, according to the results of our ongoing study (unpublished), more coarse particles were dissolved in the $\alpha$-Al matrix with increase in single-pass deformation and forging passes, but to a much lesser extent than that of the MDF temperature.

The characteristics of undissolved $Al_2Cu$ particles and precipitates were summarized in Figure 12f,g. It is well known that solubility of $Al_2Cu$ particles is related to the particle size, that is, $\ln[C_\alpha(r)/C_\alpha(\infty)] = 2\gamma_{\alpha\beta}V_{Cu}/kTr$ [30]. Here, $C_\alpha(r)$ and $C_\alpha(\infty)$ are the solubility of solute atoms in $\alpha$-Al matrix when the curvature radius of $\alpha/Al_2Cu$ interface is r and $\infty$, respectively. $\gamma_{\alpha\beta}$ is the interface energy, $V_{Cu}$ is the volume fraction of solute atoms, $k$ is the Boltzmann constant, and $T$ is the temperature in centigrade. If the size of $Al_2Cu$ particles is smaller, the curvature radius of $\alpha/Al_2Cu$ interface is smaller but the $C_\alpha(r)$ is larger, resulting in the concentration gradient of $\alpha$-Al matrix between $Al_2Cu$ particles with different sizes. In order to maintain the concentration balance at $\alpha/Al_2Cu$ interface, Cu atoms will diffuse from

the side of smaller particles to the other side of larger particles. Finally, the fine Al₂Cu fragments were fully dissolved and coarse Al₂Cu particles were partially dissolved into the Al matrix. Thus, the solid-solution strengthening contributed to improve the UTS/YS/EL values, as shown in Figure 9. Subsequently, a number of quench-induced equilibrium particles preferentially nucleated along the grain boundaries during water quenching, as shown in Figures 7 and 8. Then, the precipitated phases θ'' and θ' were separated out by the aging treatment. Thus, the precipitation hardening contributed to further increase of the UTS/YS values, whereas areas of strain concentration were formed at the interface of θ' and matrix, leading to decrease of elongation. In other words, the alloys revealed the highest strength and the lowest elongation in the peak aged treatment, as shown in Figure 9.

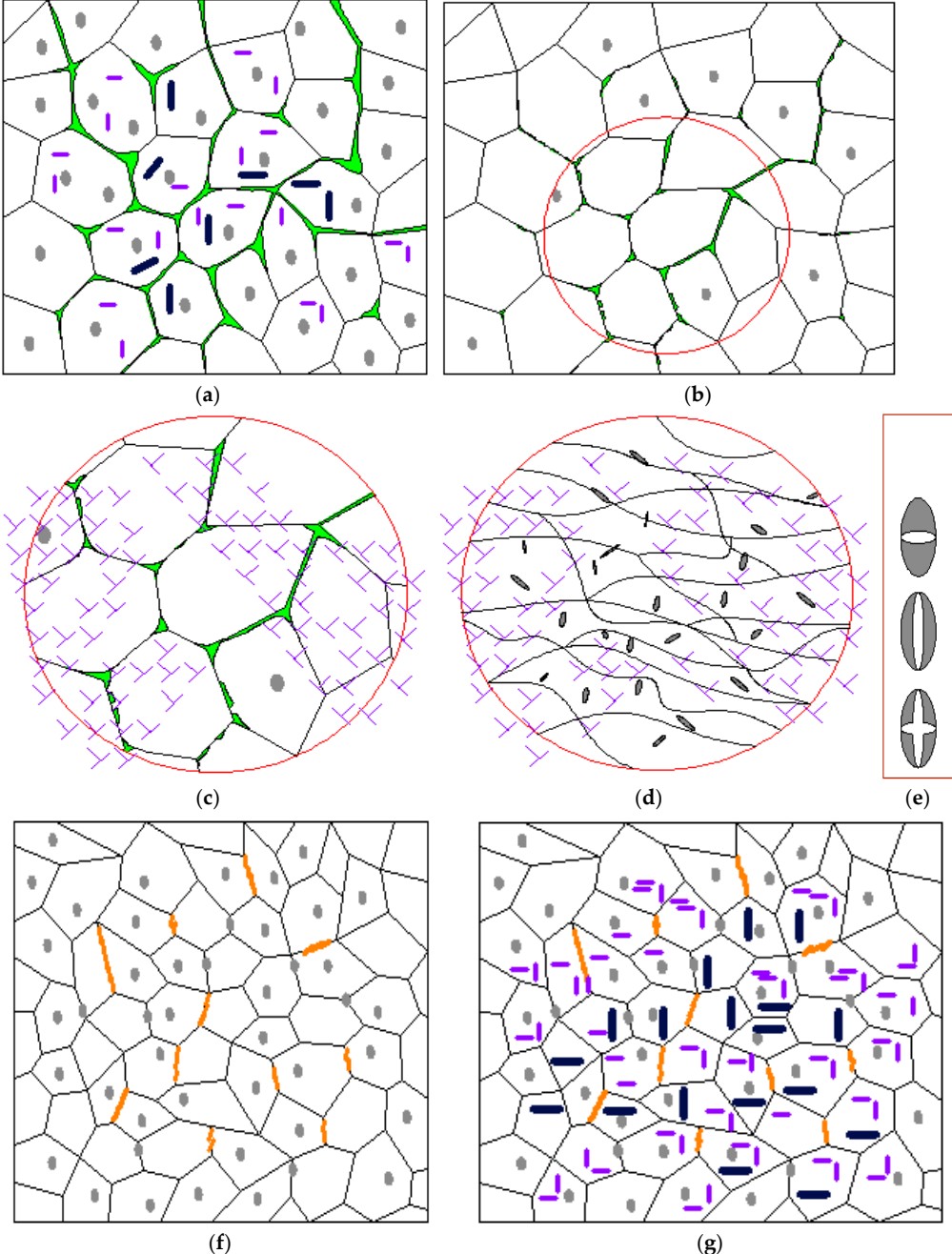

**Figure 12.** The continuous evolution of the second-phase particles undergoing different processes: (**a**) Ingot; (**b**) homogenization; (**c**–**e**) MDF; (**f**) solution treatment followed by water quenching; (**g**) peak aging treatment.

### 4.2. Effect of the Residual Second-Phase Particles on Tensile Fracture Behavior of 2219 Al-Cu Alloys

Since the microscopic characteristics of different 2219 Al-Cu alloys are the same except for the residual coarse phases, the observed differences in tensile fracture behavior can be attributed to the evolution of the residual coarse phases presented in the microstructure. During the complex TMT process, Al$_2$Cu particles exhibited an elongated shape or were spheriodized due to the interactions with fragmentation, dissolution, and diffusion, whereas, for Al$_7$Cu$_2$Fe/Al$_7$Cu$_2$(Fe, Mn) particles, the main mechanism was fragmentation. Therefore, the primary coarse particles Al$_2$Cu and Al$_7$Cu$_2$Fe/Al$_7$Cu$_2$(Fe, Mn), with multidirectional crack initiation, were retained in the alloys. So, except for the inter-granular fracture, dehiscence on primary coarse particles and decohesion at the particles-matrix interface could be the intrinsic fracture features during the room tensile process.

It is well known that the primary brittle particles are the initiator for cracks, as shown in Figure 13. One of the reasons is that they were easy to break under external service loading. Another reason is that they were broken easily when the bulk sample was subjected to large strains during MDF and thus the constituents are apt to cleave along different direction. Liu et al. [31] investigated the influences of multiscale-sized second-phase particles on ductility of Al-Cu-Mg(or Si) alloy and they found that constituents had the greatest detrimental effect on fracture toughness. For simplicity, if one neglects the dispersoids and precipitates and then assumes the Al-Cu alloys as composites including only the 'equivalent' $\alpha$-Al matrix and constituents, the tensile stress of composite, $\sigma_s$, is defined as

$$\sigma_s = f_c\sigma_c + (1 - f_c)\sigma_\alpha \tag{1}$$

where $f_c$ is the volume fraction of constituents, $\sigma_c$ and $\sigma_a$ are stresses suffered by constituents and $\alpha$-Al matrix, respectively. Due to the small value of $f_c$, $\sigma_c$ increases quickly to reach the fracture strength of constituents ($\sigma_c^*$) with a small increase of $\sigma_s$. Here, the dependence of $\sigma_c^*$ on the characteristics and size of constituents could be described as

$$\sigma_c^* = \left(\frac{6E\gamma}{K^2d}\right)^{\frac{1}{2}} \tag{2}$$

in which $E$ is the weighted mean elasticity modulus of particles and matrix (GPa), $\gamma$ is the surface energy of constituents, $K$ is the stress concentration factor, $d$ is the diameter of particles (m). $\sigma_c^*$ is in proportion to $d^{-1/2}$. For example, $\sigma_c^*$ of constituents in 2024 Al alloy degraded from 740 to 540 MPa with an increase in particles diameter and further decreased to 437 MPa with a diameter of 8.4 μm [32].

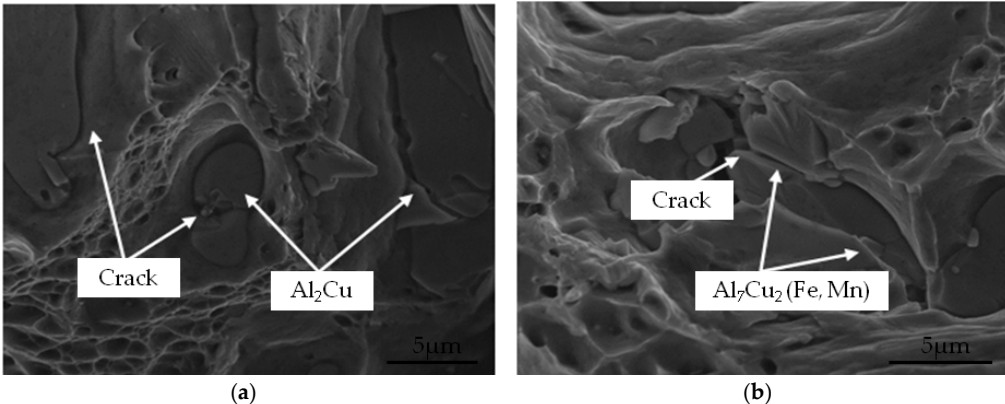

                             (**a**)                                        (**b**)

**Figure 13.** SEM morphology of residual constituents and cracks on fracture surfaces of 2219 aluminum alloy: (**a**) Cracks on Al$_2$Cu phase and coarse particles/Al matrix interfaces; (**b**) cracks appearing along Al$_7$Cu$_2$(Fe, Mn) phase.

It is clearly seen from the Equations (1) and (2) that more constituents or constituents with larger size were easy to crack. In contrast, the smaller constituents or spherical constituents were less prone to cleave, as shown in Figure 10b–d. In addition, the Fe-rich particles with a higher stress concentration

factor, which are considered as preferred sites for crack initiation, are subjected to a higher levels of stress concentration. So, the sharp edges of the Fe-rich particles became crack initiations and then cracks propagated along themselves, as shown in Figure 13b.

The $Al_2Cu$ particles are apt to cleave during MDF. So, an equation was employed by Griffith [32] to relate the tensile stress to the length of cracks, C, as

$$\sigma_c = (\frac{2E\gamma}{\pi C})^{\frac{1}{2}}$$

(3)

Equation (3) clearly shows that the larger crack at particles, the lower tensile stress suffered by constituents. However, the crack propagation mainly depends on the direction of crack initiation. When the loading direction is parallel to crack initiation, the crack does not extend. However, when the loading direction is vertical to crack initiation, there is an obvious crack propagation, as shown in Figures 12e and 13a.

In addition, Figure 13 clearly shows that the constituents are apt to separate from the matrix because of the weak adhesion at the phase-matrix interface. The tensile stress ($\sigma$) for the decohesion at the second-phase particles/matrix interface could be given by

$$\sigma = \frac{1}{K}(\frac{E\gamma}{d})^{\frac{1}{2}} + \frac{\sigma_a}{K}(\frac{\Delta V}{V})^{\frac{1}{2}}$$

(4)

where $\sigma$ is the tensile stress for separation of the second-phase particles/matrix (MPa), $V$ is the volume of particles ($m^3$), $\Delta V$ is volume increment of the deformed matrix around particles ($m^3$).

Equation (4) clearly shows that the constituents with larger size and stress concentration factor are more prone to separate from the matrix. A large number of edge dislocations are formed along the slip planes during the tensile process and these dislocations could create slip, and finally pile up at phase-matrix interface due to the pinning effect of the $Al_2Cu$ or $Al_7Cu_2Fe/Al_7Cu_2$(Fe, Mn) particles, which can create tensile stress field. Once the stress is greater than the bonding strength of the phase-matrix interface, particles are found to decohere from the matrix and thus cracks can be formed. In addition, as shown in Equation (4), the larger the second phase particles, the lower cohesion at the phase-matrix interface, which is in line with the fact that coarse particles will promote crack growth, whereas particles with smaller sizes exhibit excellent debonding resistance and limit the crack propagation.

Overall, multi-directional cracks, produced on the coarse $Al_2Cu$ particles during MDF and decohesion at $Al_2Cu$ particles-matrix interface, were the main reason for tensile fracture failure of 2219 aluminum alloys. In addition, the low strength impurity $Al_7Cu_2Fe/Al_7Cu_2$(Fe, Mn) particles could produce brittle fracture, which tends to accelerate fracture failure. However, in order to retain the welding property of 2219 aluminum alloy and reduce costs, low melting point Al-Cu eutectic phases and impurity phases inevitably exist. Therefore, it is necessary to adjust the content of the main element Cu and control the content of the impurity element Fe.

## 5. Conclusions

In the present study, the evolution of the second-phase particles during as-cast, homogenization, MDF, and solution treatment was conducted on the 2219 Al-*x*Cu alloys. The effect of the second-phase particles on tensile fracture behavior was investigated and the main conclusions were summarized as follows:

(1) The main constituents in as-cast 2219Al-Cu alloys were $Al_2Cu$ and $Al_7Cu_2Fe/Al_7Cu_2$(Fe, Mn). The spherical constituents were distributed uniformly inside the grains while the lamellar eutectic phase segregated seriously along the grain boundaries. The aggregation of $Al_2Cu$ constituents increased with increase in Cu content. For all the homogenized samples, the width of eutectic networks became smaller.

(2) During MDF, the refinement mechanisms of $Al_2Cu$ phase were fragmentation, dissolution, and diffusion, while the refinement mechanism of the $Al_7Cu_2Fe/Al_7Cu_2$(Fe, Mn) phase was only fragmentation. Consequently, the $Al_2Cu$ particles were fragmented into smaller particles or exhibited an elongated shape, while the needle-like $Al_7Cu_2Fe/Al_7Cu_2$(Fe, Mn) particles were fragmented into short rod-like particles. Meanwhile, visible cracks existed on primary coarse particles of the large size.

(3) During the solution treatment, fine $Al_2Cu$ particles were fully dissolved into Al matrix and coarse $Al_2Cu$ particles were partially dissolved. The amount of residual second-phase particles clearly decreased and their size became smaller. A number of quench-induced equilibrium particles preferentially nucleated along the grain boundaries during the water quenching.

(4) The UTS, YS, and EL of all the samples increased significantly after solution treatment followed by water quenching. These improved mechanical properties may be caused by the complex TMT process-induced second-phase particles refinement. The original microcracks propagation on coarse constituent particles and microvoids nucleation at the interface of the second-phase particles and matrix played a major role in tensile fracture failure. Unluckily, elongations clearly decreased at peak aged conditions due to coarsen of the precipitates.

**Author Contributions:** D.X. conceived and conducted writing-original draft preparation. Y.C. performed the experiments. K.C. and S.C. derived the theoretical framework. All authors have read and agreed to the published version of the manuscript.

**Funding:** This research was funded by the National Natural Science Foundation of China (U1637601), the National Key Research and Development Program of China (2016YFB0300801), the Major Research Equipment Development Projects of National Natural Science Foundation of China (51327902), the Opening Project of Guangxi Colleges and Universities Key Laboratory of robot & welding (JQR2018ZR02) and the Scientific Research Project of Guangxi Education Department (KY2015ZD141).

**Conflicts of Interest:** The authors declare no conflict of interest.

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
