# Peer review of "Evolution of the Second-Phase Particles and Their Effect on Tensile Fracture Behavior of 2219 Al-xCu Alloys"

_metals, doi:10.3390/met10020197_

Round 1
Reviewer 1 Report
The research is well conducted; the following small details should be corrected.
Line 37: I dont understand the meaning of the words "the same bellow" in the sentence in brackets. Revise this, please. Line 110: The title of epigraph 3 is alone at the bottom of the page. It will be better be at the beggining of page 4 with the rest of the epigraph. Section 3.1: It will be better to use another expression instead of "grain interior", as for instance "grain core" that sounds more scientific. Figure 5 (d): A detail of the cracks in a higer nagnification could be included, if possible. Section 3.3: The standard/procesire used for the grain size should be indicated. Figure 9: Explain what the lines at the top of the bars are. Error markers?Author Response
Point 1: Line 37: I dont understand the meaning of the words "the same bellow" in the sentence in brackets. Revise this, please.
Response 1: It has been revised.
Point 2: Line 110: The title of epigraph 3 is alone at the bottom of the page. It will be better be at the beginning of page 4 with the rest of the epigraph.
Response 2: It has been revised.
Point 3: Section 3.1: It will be better to use another expression instead of "grain interior", as for instance "grain core" that sounds more scientific.
Response 3: The expression "in grain interior" has been replaced by "inside the grains".
Point 4: Figure 5 (d): A detail of the cracks in a higher magnification could be included, if possible.
Response 4: I am so sorry for that I can’t provide another higher magnification for the cracks on the second particles. Because I spend the Spring Festival in my hometown while the pictures, related to this paper, stay in school. However, it is clearly seen from figure 5d and figure 13a that visible cracks on the second particles were formed. If necessary, could I upload it when I go back to school?
Point 5: Section 3.3: The standard/procesire used for the grain size should be indicated.
Response 5: The average grain size was measured by a linear intercept method (ASTM E112-10).
Point 6: Figure 9: Explain what the lines at the top of the bars are. Error markers?
Response 6: They are error bars.
Reviewer 2 Report
Please, consult the English language editor.
How is inter-granular fracture ductile and trans-granular brittle? This is mentioned in Abstract (lines 26-27) and on page 11 - line 228? Shouldn't it be opposite?
Line 330: What is "the room tensile process"?
Some abbreviations are not explained. You cannot use abbreviation "MDF" in Abstract without explanation what does it stand for.
Font size and type in equations are not the same as in the rest of the text.
Equations are parts of the text and should be followed by the punctuation mark in the same line (comma or full-stop).
What is "Ktr" in line 306, page 14?
"MPa" stands for "Mega Pascal", unit is named after a scientist Blaise Pascal and it must be written properly, NOT "Mpa" .
The technical text should be written in the neutral form, i.e. in the third person, not the first.
The scanned pages of the article with suggested corrections are enclosed.

Author Response
Point 1: How is inter-granular fracture ductile and trans-granular brittle? This is mentioned in Abstract (lines 26-27) and on page 11 - line 228? Shouldn't it be opposite?
Response 1: For 5.56% content of Cu, fine spheroidized particles were evenly distributed inside the grains. The occurrence of inter-granular fracture might have resulted from the stress concentration caused by the grain boundary. With increase in Cu content, spheroidized particles were gradually reduced, while the coarse particles exhibited an elongated shape. The occurrence of trans-granular fracture might have resulted from the stress concentration caused by the coarse particles. Hence, the reduction in the brittle coarsening particles can improve the ductility of 2219 Al alloy, reducing the formation of cracks during deformation process.Thus, the fracture mode was diverted from a typical inter-granular fracture to a mainly trans-granular fracture with increase in Cu content from 5.56% to 6.52%.
Point 2: 330: What is "the room tensile process"?
Response 2: The geometry of tensile specimen was designed according to the standard of GB/T228-2002 and all the tests were carried out using an Instron 3369 electronic universal testing machine at room temperature with a constant speed of 2mm/min.
Point 3: abbreviations are not explained. You cannot use abbreviation "MDF" in Abstract without explanation what does it stand for.
Response 3: It has been revised.
Point 4: Font size and type in equations are not the same as in the rest of the text.
Response 3: It has been revised.
Point 5: Equations are parts of the text and should be followed by the punctuation mark in the same line (comma or full-stop).
Response 5: It has been revised.
Point 6: What is "Ktr" in line 306, page 14?
Response 6: k is the Boltzmann constant, T is the temperature in centigrade and r is the curvature radius of α/Al2Cu interface. All the meanings of symbols have been added to this paper.
Point 7: "MPa" stands for "Mega Pascal", unit is named after a scientist Blaise Pascal and it must be written properly, NOT "Mpa" .
Response 7: It has been revised.
Point 8: The technical text should be written in the neutral form, i.e. in the third person, not the first.
Response 8: It has been revised.
Reviewer 3 Report
In the present paper a deep study in presented in order to correlate microstructure and mechanical properties of Al-xCu alloys. In my opinion the paper is worth for publication after minor revisions:
1) abstract: please use subscript where necessary (e.g. Al2Cu) and add acronym significate (MDF)
2)plase indicate how the compositions in table 1 have been measured or derived.
Author Response
Point 1: abstract: please use subscript where necessary (e.g. Al2Cu) and add acronym significate (MDF)
Response 1: It has been revised.
Point 2: please indicate how the compositions in table 1 have been measured or derived.
Response 2: The chemical compositions were measured by inductively coupled plasma optical emission spectroscopy.